# Preparation and Characterization of Mono- and Biphasic Ca_1−x_Ag_x_HPO_4_·nH_2_O Compounds for Biomedical Applications

**DOI:** 10.3390/biomimetics8070547

**Published:** 2023-11-14

**Authors:** Fahad Abdulaziz, Khalil Issa, Mohammed Alyami, Satam Alotibi, Abdulaziz A. Alanazi, Taha Abdel Mohaymen Taha, Asma M. E. Saad, Gehan A. Hammouda, Nagat Hamad, Mazen Alshaaer

**Affiliations:** 1Department of Chemistry, College of Science, University of Ha’il, Ha’il 81451, Saudi Arabia; fah.alanazi@uoh.edu.sa; 2Orthopedics Unit, Faculty of Medicine and Health Sciences, An-Najah National University, Nablus 00972, Palestine; k.issa@najah.edu; 3Department of Physics, College of Science and Humanities in Al-Kharj, Prince Sattam bin Abdulaziz University, Al-Kharj 11942, Saudi Arabia; m.alyami@psau.edu.sa (M.A.); sf.alotibi@psau.edu.sa (S.A.); aasmasaada@gmail.com (A.M.E.S.); n.hamad@psau.edu.sa (N.H.); 4Department of Chemistry, College of Science and Humanities in Al-Kharj, Prince Sattam bin Abdulaziz University, Al-Kharj 11942, Saudi Arabia; abdulaziz.alanazi@psau.edu.sa (A.A.A.); g.hammouda@psau.edu.sa (G.A.H.); 5Physics Department, College of Science, Jouf University, P.O. Box 2014, Sakaka 72388, Saudi Arabia; themaida@ju.edu.sa; 6Physics and Engineering Mathematics Department, Faculty of Electronic Engineering, Menoufia University, Menouf 32952, Egypt; 7Department Mechanics of Materials and Constructions (MEMC), Vrije Universiteit Brussels (VUB), Pleinlaan 2, 1050 Brussels, Belgium

**Keywords:** brushite, silver phosphate, crystal structure, phase composition, Rietveld refinement

## Abstract

This study aimed to explore the effects of the full-scale replacement (up to 100%) of Ca^2+^ ions with Ag^1+^ ions in the structure of brushite (CaHPO_4_·2H_2_O). This substitution has potential benefits for producing monophasic and biphasic Ca_1−x_Ag_x_HPO_4_·nH_2_O compounds. To prepare the starting solutions, (NH_4_)_2_HPO_4_, Ca(NO_3_)_2_·4H_2_O, and AgNO_3_ at different concentrations were used. The results showed that when the Ag/Ca molar ratio was below 0.25, partial substitution of Ca with Ag reduced the size of the unit cell of brushite. As the Ag/Ca molar ratio increased to 4, a compound with both monoclinic CaHPO_4_·2H_2_O and cubic nanostructured Ag_3_PO_4_ phases formed. There was a nearly linear relationship between the Ag ion ratio in the starting solutions and the wt% precipitation of the Ag_3_PO_4_ phase in the resulting compound. Moreover, when the Ag/Ca molar ratio exceeded 4, a single-phase Ag_3_PO_4_ compound formed. Hence, adjusting the Ag/Ca ratio in the starting solution allows the production of biomaterials with customized properties. In summary, this study introduces a novel synthesis method for the mono- and biphasic Ca_1−x_Ag_x_HPO_4_·nH_2_O compounds brushite and silver phosphate. The preparation of these phases in a one-pot synthesis with controlled phase composition resulted in the enhancement of existing bone cement formulations by allowing better mixing of the starting ingredients.

## 1. Introduction

Currently, there is a strong demand for improved bone substitutes and implants to treat various bone defects in the body [1,2] stemming from causes like osteoporosis, surgeries, and fractures. Traditionally, autografts are the common solution for bone replacement [2]. However, conventional bone implants and substitutes pose challenges, such as a high infection risk, limited graft materials, and complications [3,4]. Moreover, bone marrow treatment faces limitations due to donor shortages and safety concerns [5,6]. Implant-related infections are a significant issue with serious clinical and socioeconomic consequences [7,8]. Bacterial biofilm formation on inert surfaces and nearby tissues is a major problem, reducing susceptibility to host defenses and antimicrobial treatments [9]. Current remedies involve surgery, antibiotics, and even implant removal, leading to more surgeries, extended hospital stays, and increased mortality [10]. To address these challenges, synthetic bone materials have emerged as a safer alternative to autografts, offering reduced infection risks and complications. There is also a rising need for clinically viable implants that can naturally degrade in the body.

Biodegradable materials are promising as bone substitutes because they can be absorbed and replaced by natural bone tissue over time, eliminating the need for additional surgeries. These materials possess osteogenic, osteoinductive, and osteoconductive properties. However, a significant obstacle in using biodegradable materials for bone regeneration is their integration and vascularization within the body [11,12]. Calcium phosphate (CaP) plays a crucial role in various fields, including medicine, biology, engineering, and environmental science, due to its structural characteristics [13]. CaPs are advantageous for medical applications because they are biocompatible, low in toxicity, highly bioactive, and similar to natural mineral phases in bone tissue [1,6]. Currently, CaPs are used in advanced biocements and bioceramics for dental and medical interventions [11,14].

In supersaturated solutions, various CaP phases can form, such as amorphous calcium phosphate (ACP) or hydroxyapatite (HA), brushite (DCPD), monetite (DCPA), β-tricalcium phosphate (β-TCP), and octacalcium phosphate (OCP) [13,14,15,16,17,18]. These phases vary in solubility and crystalline structures. Calcium phosphate phases can spontaneously form in aqueous supersaturated solutions or with substrate assistance through seeded crystal growth [13]. Dicalcium phosphate dihydrate (DCPD, CaHPO_4_·2H_2_O), known as brushite, is crucial, serving as a precursor for bioceramics, bone cement, and hydroxyapatite (HA) [12,19]. It remains stable at pH levels of 4.0–6.5 and low temperatures [20] and can be reabsorbed by the body, supporting bone tissue remodeling, as it is typically metastable at a pH of around 7.4 [21]. 

Brushite has been utilized as a precursor for other CaP phases, such as HA and OCP. This mineral (phase) crystallizes in a monoclinic structure (space group Ia) with lattice parameters a = 5.81, b = 15.18, c = 6.24, and β = 116.42° [3]. It exhibits a layered structure, with corrugated sheets of CaHPO_4_ and two water molecules alternating along the [010] direction. The octahedral is represented by every Ca^2+^ ion being surrounded by six phosphate oxygen molecules from four phosphoryl groups (HPO_4_)^2−^ and two oxygen molecules from two water molecules [6]. The monoclinic structure of these calcium ions allows protons to migrate toward the electrode–electrolyte interface through their interaction with the oxygen atom of the phosphate group and water molecules. Moreover, the characteristic features of brushite, like high porosity, good biocompatibility, and high adsorption capacity, make it a suitable candidate material for biomedical applications.

Unfortunately, bone implants can be vulnerable to bacterial colonization in environments with reduced blood supply and local immunodeficiency, leading to postoperative infections that may require implant removal [22,23]. Metal ions, including Ag^+^, Cu^2+^, and Zn^2+^, are recognized as antimicrobial agents in CaPs [24,25,26]. Ag+ is particularly known for its antibacterial properties against various microorganisms, including antibiotic-resistant strains [27], demonstrating high antimicrobial activity with relatively low cytotoxicity [28,29]. Dissolved Ag ions in the human body exhibit antibacterial effects through multiple mechanisms, including deactivation of electron donor groups, membrane interaction, and the generation of reactive oxygen species (ROS) and DNA damage within bacterial cells [30,31]. Consequently, the addition of Ag to CaPs has been explored in several studies [32]. However, high Ag ion concentrations in body fluids can be toxic to certain tissues [33], prompting the establishment of a recommended exposure limit (REL) for soluble Ag compounds and Ag metal dust by the National Institute for Occupational Safety and Health (NIOSH) [34].

In previous [35,36,37,38,39] studies, CaPs like HA and TCP have been utilized to synthesize biomaterials that have antibacterial properties and no cytotoxicity. It was found that Ag-containing CaPs are safe and can prevent infection for long-term in vivo applications. The production of silver phosphate and calcium phosphates was carried out separately with the use of a precipitation method [36]. Subsequently, the two phases were incorporated into a resin in mass fractions below 30% for antibacterial and biomedical purposes. One of the advanced applications of silver phosphate/CaP compounds is silver-loaded calcium phosphate bone cement with CaP phases such as brushite, HA, and TCP [37]. The molar ratio of Ag/Ca used to prepare those cements was 5%. In addition to CaPs, pure Ag_3_PO_4_ powder was introduced as a precursor of bone cement in that study. Silver phosphate showed significant effects on the morphology, setting time, and phase composition of the final cements. Loaded silver ions extended the final setting time of the bone cement by approximately 25%. The XRD indicated a change in the cement’s crystalline composition, with peaks of whitlockite visible in samples loaded with silver. According to FTIR spectroscopy, the phase of pyrophosphate was present. It has also been observed that implant-associated infections can be eliminated effectively with silver-loaded calcium phosphate bone cement [38]. The antibacterial capability and biomechanical properties of end products can be enhanced by adding silver ions to calcium phosphate bone cement, as shown in previous studies [37,38]. In summary, silver-loaded calcium phosphate bone cements are characterized by high cytocompatibility, good injectability, and good interdigitating in spongy bone, with sustainable antibacterial effects. Therefore, these cements bear great potential for the treatment of bone infections or implant-associated infections.

Both brushite (CaHPO_4_·2H_2_O) and silver phosphate (Ag_3_PO_4_) play significant roles in biomedical applications and complement each other. Nonetheless, careful consideration of the Ag content in these compounds is crucial to avoid adverse effects. Therefore, when developing bone substitutes and implants, it is vital to balance factors like biocompatibility, bioactivity, osteogenic properties, and potential antimicrobial effects. Calcium phosphate (CaP) materials, particularly dicalcium phosphate dihydrate (DCPD or brushite), hold promise in bone regeneration due to their biocompatibility and resemblance to natural bone tissue. Introducing antimicrobial agents like silver ions (Ag+) into CaP materials can help prevent implant-related infections. Nevertheless, it is imperative to carefully control the concentration of silver ions to avoid detrimental effects. 

According to the above, there has not yet been a full study of the effects of gradual full-scale (up to 100%) silver replacement for calcium in brushite, with only the effect of doping brushite with a lower percentage of silver ions, less than 5%, having been investigated. In addition, no compound containing both silver phosphate and calcium phosphate has ever been synthesized in specific ratios, despite the importance of combining both phases in biomedical applications. However, both phases have been prepared separately [37] and then mixed. Therefore, this research work’s novelty lies in its study of the effect of full-scale substitution and replacement (up to 100%) of Ca ions in brushite with Ag ions. At each level of ion replacement, the phase composition, thermal properties, and crystal morphology of the produced powders were investigated. The findings of this research fill an important research gap and may prove valuable for mineral synthesis to derive novel (Ag and Ca) phosphate-based precursors of biomaterials with unique bioactive and antibacterial characteristics. Synthesized products and minerals have great potential in bone cement and bioceramic synthesis, as well as have a variety of applications in the bone tissue engineering and pharmaceutical industries.

## 2. Materials and Methods

### 2.1. Materials 

This study used chemicals obtained from different suppliers. Diammonium hydrogen phosphate ((NH_4_)_2_HPO_4_) was sourced from Techno Pharmchem in Delhi, India. Calcium nitrate tetrahydrate (Ca(NO_3_)_2_·4H_2_O) and silver nitrate hexahydrate (AgNO_3_·6H_2_O) were purchased from LOBA Chemie in Mumbai, India. To ensure their purity, we prepared distilled water with a low conductivity of 0.055 µS/cm using the PURELAB option-Q purification system from ELGA in the UK. To ensure precise measurements, we used an EX324N digital analytical balance from OHAUS in Parsippany, NJ, USA, and an ISOTEMP magnetic stirrer from Fisher Scientific in China, as needed.

### 2.2. Synthesis of Ca_1−x_Ag_x_HPO_4_·nH_2_O Compounds

Seven different compounds of Ca_1−x_Ag_x_HPO_4_·nH_2_O were synthesized at room temperature (RT) using solutions of (NH4)_2_HPO_4_, Ca(NO_3_)_2_·4H_2_O, and AgNO_3_ with specific molar proportions, as shown in Table 1, and Figure 1.

To prepare pure brushite (referred to as BAg0 in Table 1), 100 mL of the Ca(NO_3_)_2_·4H_2_O solution was mixed with the (NH_4_)_2_HPO_4_ solution at a controlled flow rate of approximately 2 mL/min. This mixing process was conducted using a glass funnel equipped with a stopcock while maintaining a stirring speed of 450 rpm until achieving a 1 Ca/P molar ratio, which took approximately one hour. The resulting solution was stirred at RT for 60 min to ensure thorough mixing. Ammonia (~15 moL/L, Labochemie, Mumbai, India) was added to maintain the pH level within the range of 6–6.5. The resulting white precipitate was separated using a Büchner funnel and qualitative filter paper (45 µm, Ø 12 cm, Double Rings, China). The filter cake underwent three washes with deionized water and ethanol to prevent clumping [40,41,42]. Finally, the sample was dried in an oven (set at 40 °C) for 7 days on a watch glass (ED53/E2, Binder, Tuttlingen, Germany) [18].

For compounds BAg2, BAg4, BAg5, BAg6, and BAg10, we combined the Ca(NO_3_)_2_·4H_2_O and AgNO_3_ solutions in the molar ratios outlined in Table 1. Following a procedure similar to the one described earlier, 100 mL of the resulting solution was introduced into 100 mL of the (NH_4_)_2_HPO_4_ solution at a controlled flow rate of approximately 2 mL/min. The same approach was used to prepare BAg10 but using a mixture of (NH_4_)_2_HPO_4_ and AgNO_3_.

### 2.3. Characterization Techniques

Qualitative mineralogical analysis was conducted on the BAg0–BAg10 samples using a Shimadzu XRD diffractometer-6000 from Kyoto, Japan. The analysis involved using a cobalt tube and scanning in the 2-theta range of 10–60° at a scan rate of 2°/min. For phase and crystal analysis, Match! software (version 3.15, Crystal Impact, Bonn, Germany) was used to process powder XRD diffraction data.

To obtain a comprehensive view of the product morphology, scanning electron microscopy with an Inspect F50 instrument was used (FEI Company, Eindhoven, The Netherlands). Surface chemistry and in-depth sample analysis were evaluated using an XPS system, specifically the Thermo K Alpha spectrometer from the USA, which conducted elemental X-ray photoelectron spectroscopy.

Additionally, a thermogravimetric (TG) analyzer (Netzsch, Waldkraiburg, Germany, TG 209 F1 Libra) was used to measure the mass loss (approximately 100 mg) of each product as they were heated from 40 to 750 °C, in increments of 5 °C/min, under a helium atmosphere.

To determine the silver (Ag) content of the synthesized compounds, inductively coupled plasma optical emission spectrometry (ICP-OES) using the Thermo Fisher Scientific iCAP 7000 series instrument was used.

## 3. Results and Discussion

### 3.1. Mineralogical and Microstructural Analysis

In Figure 2, we present XRD scan analyses of various samples, including pure Ca-HPO_4_·2H_2_O (Ag0) and pure Ag_3_PO_4_ (Ag10). The qualitative mineralogical analysis confirmed the formation of pure brushite (BAg0) by combining solutions of Ca(NO_3_)_2_·4H_2_O and (NH_4_)_2_HPO_4_ with a Ca to P molar ratio of 1:1, as detailed in Table 1. The crystals showed expansion in three primary crystallographic planes with Miller indices of (020), (12 $\overset{-}{1}$), and (14 $\overset{-}{1}$), indicating a monoclinic structure [43]. Importantly, the XRD peak at 2-theta of 11.7° indicated significant crystal growth along the (020) plane [43].

As the Ag/Ca molar ratios increased from 0.25 to 4, biphasic compounds of CaHPO_4_·2H_2_O and Ag_3_PO_4_ formed, as evidenced by the patterns of BAg2, BAg4, BAg5, BAg6, and BAg8. The prominent planes corresponding to the new phase (Ag_3_PO_4_) included (110), (200), (210), (211), (220), (310), (333), (320), and (321) [39]. When the complete replacement of Ca with Ag ions occurred in the initial solutions, a monophasic compound of Ag_3_PO_4_ (Ag10) was formed. Residual salts of ammonium phosphate were observed between 10° and 30° in the XRD pattern (Figure 2).

To delve deeper into the phase composition and unit cell parameters of the compounds, we conducted XRD scans and used Rietveld refinement and MATCH! software analysis, with the results reported in Table 2. The increase in Ag ions in the initial solutions (Table 1) corresponded to an increase in the proportion of Ag_3_PO_4_ at the expense of CaHPO_4_·2H_2_O. The weight percentage of CaHPO_4_·2H_2_O decreased while the weight percentage of Ag_3_PO_4_ increased, exhibiting an almost linear relationship (Figure 3). The crystal structures, lattice parameters, and unit cell volumes of both phases matched their respective standards [18,44,45]. The unit cell volume of CaHPO_4_·2H_2_O decreased with an increasing Ag/Ca molar ratio up to 0.67 but remained relatively stable thereafter. These variations in unit cell volume indicate that the Ag ions not only participated in Ag_3_PO_4_ precipitation but also became integrated into the CaHPO_4_·2H_2_O lattice as dopants or substitutes. A monophasic phase of Ag_3_PO_4_ (BAg10) was formed when the Ag/Ca ratio exceeded 4.

In Figure 4, the SEM images reveal monophasic CaHPO_4_·2H_2_O (BAg0) and Ag_3_PO_4_ (Ag10), as well as biphasic compounds (BAg2, BAg4, BAg5, BAg6, and BAg8), with varying Ag/Ca molar ratios. Monoclinic crystals of brushite (BAg0) with dimensions of 0.5 × 5 × 10 µm^3^ were observed, consistent with prior studies [46,47]. As the Ag/Ca molar ratio increased, nanosized cubic crystals of Ag_3_PO_4_ appeared alongside monoclinic CaHPO_4_·2H_2_O, as evident in the SEM images. The Ag_3_PO_4_ content increased with higher Ag/Ca molar ratios. Notably, BAg10 exhibited larger cubic crystals of Ag_3_PO_4_ compared with the other compounds.

The SEM analysis corroborated the XRD findings, confirming the presence of Ag_3_PO_4_ in the resulting compounds. The elemental analysis revealed an incremental trend in Ag content. These results established a clear relationship between the proportions of Ag_3_PO_4_ and the Ag content in the initial solutions (Figure 3 and Figure 4), confirming the involvement of Ag in various stages of the reactions

### 3.2. FTIR Spectrum of the Ca_1−x_Ag_x_HPO_4_·nH_2_O Compounds

In Figure 5, we present the infrared spectra (FTIR) of the compounds CaHPO_4_·2H_2_O and Ag_3_PO_4_. These FTIR results align with the XRD analysis findings. The spectrum of CaHPO_4_·2H_2_O (Ag0) displays distinct absorption bands that are characteristic of this phase [48,49]. Notably, the broad absorption peak spanning from 2400 to 3600 cm^–1^ in BAg0, BAg2, BAg4, BAg5, BAg6, and BAg8 is attributed to the O–H stretching vibration within CaHPO_4_·2H_2_O [50]. The presence of P–O–P asymmetric stretching vibrations is evidenced by the band at 983 cm^–1^, P=O stretching, while the bands at 654 and 569 cm^–1^ may be attributed to (H–O–)P=O in acid phosphates [51]. Additionally, the peak at 1643 cm^–1^ indicates the presence of water, with its intensity diminishing as the CaHPO_4_·2H_2_O content decreased in the samples (BAg2, BAg4, BAg5, BAg6, BAg8, and BAg10).

As the Ag content in the initial solutions increased from BAg2 to BAg10, new peaks emerged at 960 and 548 cm^–1^, corresponding to P–O bending vibrations in Ag_3_PO_4_. The presence of ammonium cations in the BCo10 compound was confirmed with infrared spectroscopy, revealing bands at 2840 and 1440 cm^–1^ [29,30].

### 3.3. Elemental Analysis of CaHPO_4_·2H_2_O and Ag_3_PO_4_

XPS analysis was conducted to examine the chemical states of Ag, Ca, and P in the prepared CaHPO_4_·2H_2_O and Ag_3_PO_4_ compounds, shedding light on their surface chemistry, as depicted in Figure 6. The phase composition of these compounds was notably influenced by the Ag/Ca molar ratio in the initial solution. The XPS spectra of compounds ranging from Bag2 to Bag10 revealed distinct peaks associated with the Ag 3d and Ag 3p orbitals. These peaks’ concentrations gradually increased, reaching their peak intensity at Bag10, while the intensity of the Ca 2p and Ca 2s peaks exhibited a proportional decrease with the escalating Ag/Ca ratio. Conversely, the intensity of the P 2s peak remained relatively stable, indicating that the concentrations of the P, Ca, and Ag peaks were contingent on the extent of Ca substitution with Ag and the quantities of precipitated compounds, particularly at higher Ag/Ca molar ratios.

The influence of the Ag/Ca ratio on the binding energies of Ag 3d, Ca 2s, and P 2s, along with the corresponding peaks, is evident in Figure 7A–C, respectively. Notably, for the Ag/Ca 0.25 (BAg2) molar ratio, the binding energies of the P 2s and Ca 2s peaks exhibited an increase from 438 to 442 eV and from 190 to 193 eV, respectively [3]. Furthermore, the Ca 2s and Ca 2p peaks disappeared in the BAg10 compound because no Ca ions were added to the starting solutions (see Table 1). The binding energy attributed to Ag 3d was observed at 371 eV for biphasic compounds with Ag_3_PO_4_ contents ranging from BAg2 to BAg8 [52].

Further evidence of the Ag/Ca ratio’s influence can be observed in the assessments of P 2s, as presented in Figure 7C. The appearance of Ca 2s peaks in BAg2 to BAg8 signifies the co-precipitation of CaHPO_4_·2H_2_O alongside Ag_3_PO_4_. However, these peaks vanished at higher Ag/Ca molar ratios (BAg10) due to the formation of pure Ag_3_PO_4_. In summary, the XPS findings firmly establish that an increase in the Ag/Ca molar ratio in the initial solution (BAg2–BAg8) led to the formation of a biphasic phosphate compound composed of CaHPO_4_·2H_2_O and Ag_3_PO_4_. Conversely, altering the crystal structure of the CaHPO_4_·2H_2_O and Ag_3_PO_4_ compounds necessitated raising the binding energies of the Ca 2s and P 2s peaks. As the Ag intensity rose and the Ca concentration decreased, the degree of supersaturation concerning CaHPO_4_·2H_2_O diminished. Consequently, a pure Ag_3_PO_4_ compound was obtained when the Ca availability was low (below 20%), as observed in the BAg10 compound.

The XPS results corroborate the presence of Ag in compounds ranging from BAg2 to BAg10. Notably, despite setting the Ag/Ca molar ratio to 4 in the starting solutions, no Ca ions were detected in BAg10, aligning with the XRD analysis presented in Figure 2. Lastly, the shifts in the peaks of Ca and P to higher positions, resulting from the addition of Ag in the starting solutions (BAg2–BAg10), signify alterations in the surroundings of these elements induced by the precipitation of the new phase (Ag_3_PO_4_).

### 3.4. Thermogravimetric Analysis (TGA) 

The TGA analysis results for the investigated materials (BAg0–BAg10) are presented, shedding light on the arrangement of Ca ions with six PO_4_ ions and two oxygen atoms belonging to the structural water in brushite [53]. Additionally, the presence of Ca-HPO_4_·2H_2_O was confirmed with the observation of two distinct peaks indicating mass loss during the temperature increase from 80 to 220 °C. These peaks are indicative of the release of the structural water molecules within the lattice and the adsorbed water molecules on the surface [42,54].

Based on the available evidence, it is suggested that a portion of the chemically attached water was released during the conversion of brushite to monetite (CaHPO_4_) at around 220 °C, with further conversion to calcium pyrophosphate (Ca_2_P_2_O_7_) occurring as the temperature rose to approximately 400 °C [55]. In the current study, heating pure CaHPO_4_·2H_2_O (BAg0) to 750 °C resulted in a mass loss of approximately 24.8 wt%, consistent with the expected hypothetical mass loss of 20.93 wt% [56]. In contrast, the BAg2–BAg10 samples, with increasing Ag/Ca ratios, experienced lower mass losses ranging from 5.4% to 20.1% (Figure 8).

The dehydration reaction of brushite is represented by Equation (1), while the formation of calcium pyrophosphate can be explained by Equation (2).
(1)$${CaHPO}_{4}\cdot 2H_{2}O\to {CaHPO}_{4}+2H_{2}O$$
(2)$${2CaHPO}_{4}\to {Ca}_{2}P_{2}O_{7}+H_{2}O$$

Figure 9 illustrates the mass loss rate of the compounds as a function of temperature. Previous research [42] displayed clear dehydration peaks below 200 °C, attributed to the two structural water molecules of CaHPO_4_·2H_2_O (BAg0), as shown in Figure 9A–F. The thermal stability of the monophase Ag_3_PO_4_ compound (BAg10) was also studied, as depicted in Figure 9. Weight loss below 150 °C is attributed to the release of weakly adsorbed water, while a second weight loss between 150 and 260 °C corresponds to the dehydration of pristine Ag_3_PO_4_ [57]. The gradual decrease in weight loss associated with CaHPO_4_ in the range of 470–410 °C strongly indicates the increased presence of Ag_3_PO_4_ at the expense of CaHPO_4_·2H_2_O in the resulting compounds, from BAg0 to BAg10. These findings align with the previous XRD analysis results presented in Figure 3.

## 4. Conclusions

This research work highlighted the effect of full-scale gradual replacement of Ca ions with Ag ions in CaHPO_4_·2H_2_O. This process was thoroughly examined using a series of analyses, including XRD, SEM, XPS, FTIR, and TG, focusing on various Ca_1−x_Ag_x_HPO_4_·nH_2_O compounds. This research achieved two main goals, as the first goal was to study the effect of the gradual full-scale replacement of Ag ions with Ca ions in brushite. Changes in crystal morphology, phase composition, and thermal properties were studied at each step of the replacement. The second goal was to synthesize a biphasic compound from silver phosphate and brushite for antibacterial and biomedical applications.

The results of Ca ion replacement with silver ions showed that when the Ag/Ca molar ratio in the initial solution fell below 0.25, Ag could partially substitute for Ca. With a gradual increase in the Ag/Ca molar ratio up to 4, biphasic compounds comprising CaHPO_4_·2H_2_O and Ag_3_PO_4_ appeared. The Ag_3_PO_4_ phase, characterized by a cubic crystal structure, was precipitated at an Ag/Ca molar ratio of 0.25. As the Ag/Ca molar ratio continued to rise, reaching higher supersaturation levels concerning Ag, a biphasic compound featuring monoclinic brushite and cubic Ag_3_PO_4_ was obtained. When the molar ratio of silver to calcium exceeded 4, a monophasic nanostructured Ag_3_PO_4_ precipitated. There was a nearly linear relationship between the Ag ion ratio in the starting solutions and the wt% precipitation of the Ag_3_PO_4_ phase in the resulting compound.

These findings hold significant implications for the future production of biomaterials, enabling the synthesis of materials with precise phase compositions, distinct structural characteristics, and tailored properties. This underscores the potential to achieve specific combinations of material components and geometries by controlling the Ag/Ca ratio in the initial solution.

## Figures and Tables

**Figure 1 biomimetics-08-00547-f001:**
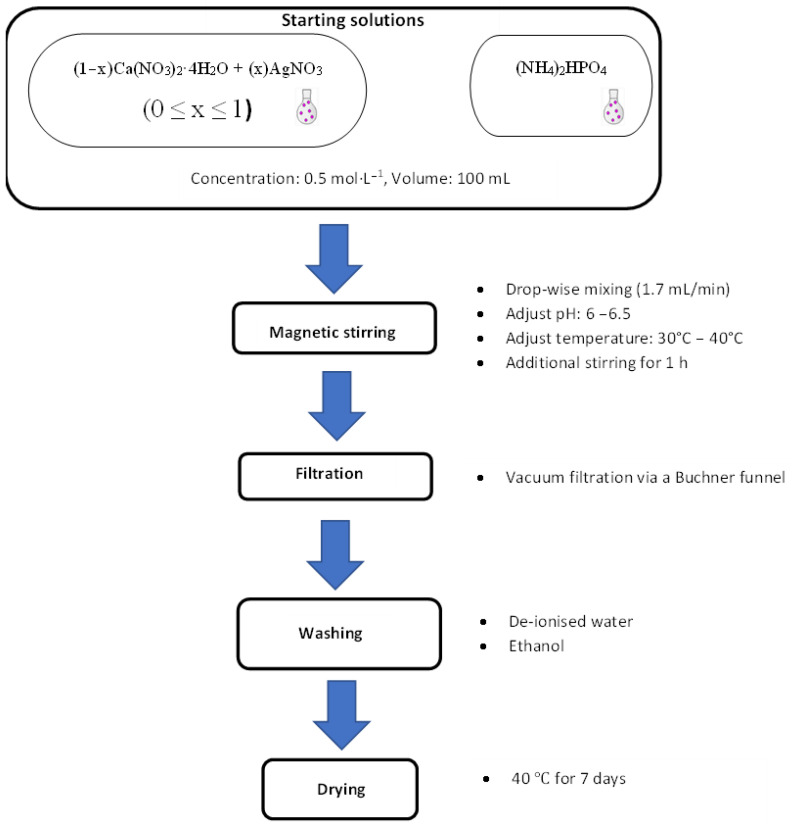
Experimental procedure for the synthesis of the Ca_1−x_Ag_x_HPO_4_·nH_2_O compounds.

**Figure 2 biomimetics-08-00547-f002:**
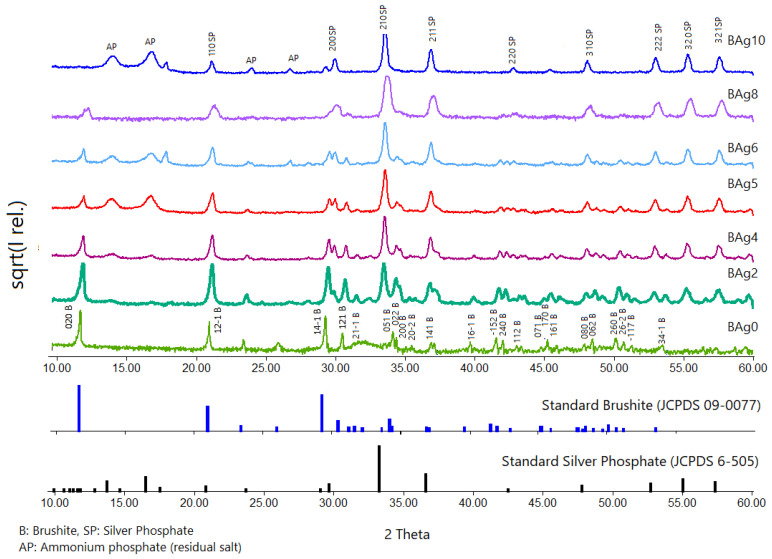
X-ray diffraction patterns of the Ca_1−x_Ag_x_HPO_4_·nH_2_O compounds produced under the conditions shown in Table 1 and Figure 1.

**Figure 3 biomimetics-08-00547-f003:**
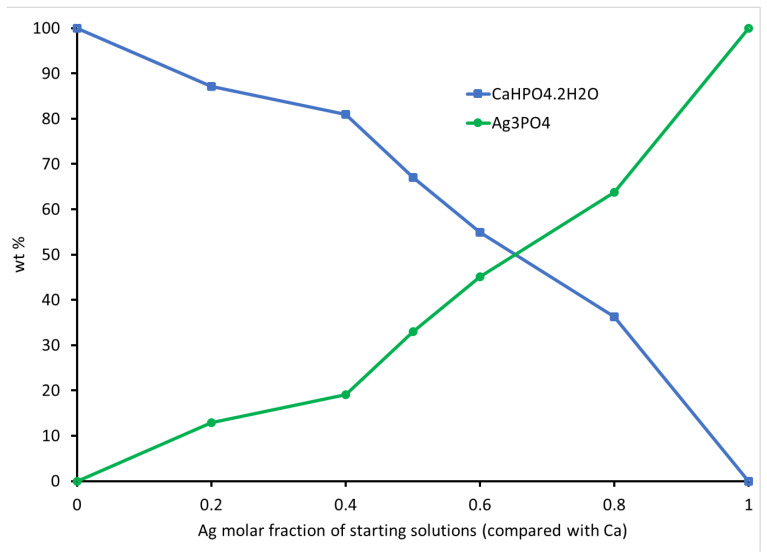
Phase composition of the produced compounds compared with the Ag molar fraction in the starting solutions.

**Figure 4 biomimetics-08-00547-f004:**
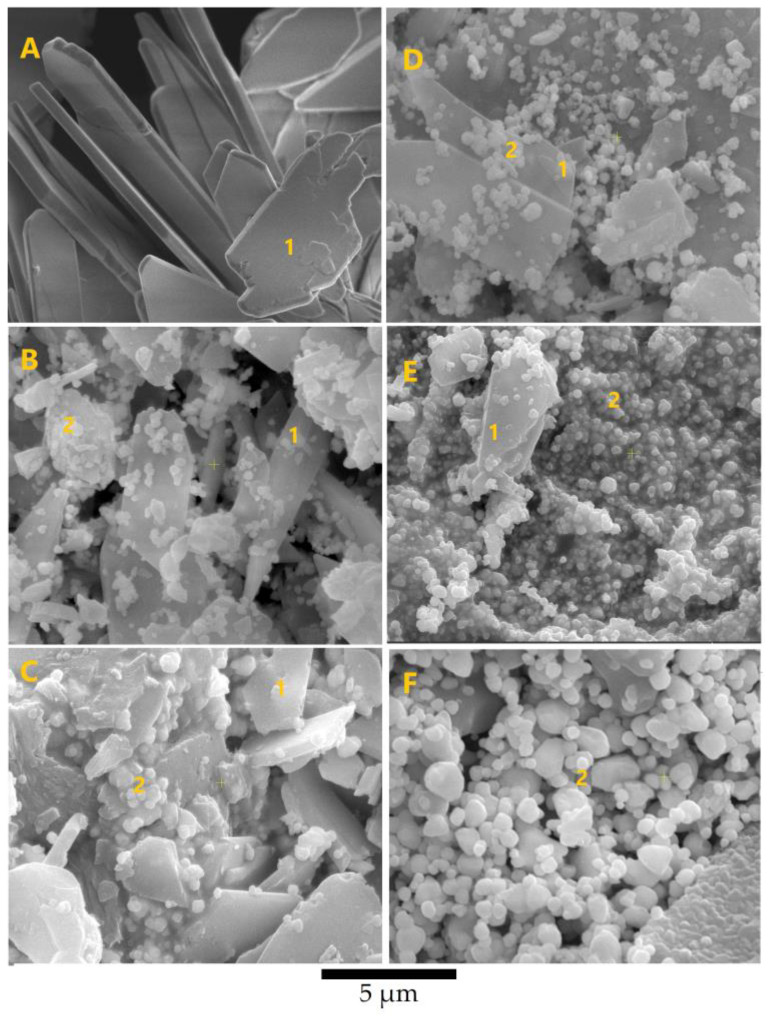
SEM images of the Ca_1−x_Ag_x_HPO_4_·nH_2_O compounds (labeled using the compound names shown in Table 1): (**A**) BAg0, (**B**) BAg4, (**C**) BAg5, (**D**) BAg6, (**E**) BAg8, and (**F**) BAg10. Point 1, CaHPO_4_·2H_2_O crystals; point 2, AgPO_4_ crystals.

**Figure 5 biomimetics-08-00547-f005:**
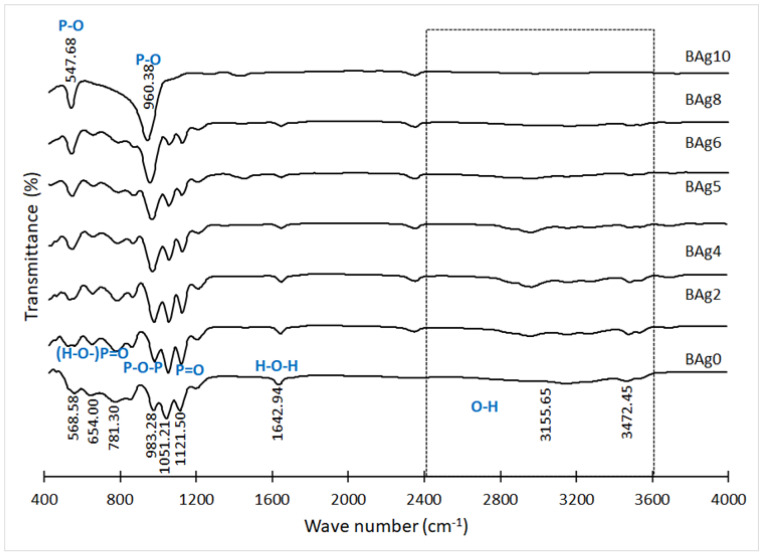
FTIR spectrum of the CaHPO_4_·2H_2_O and Ag_3_PO_4_ compounds.

**Figure 6 biomimetics-08-00547-f006:**
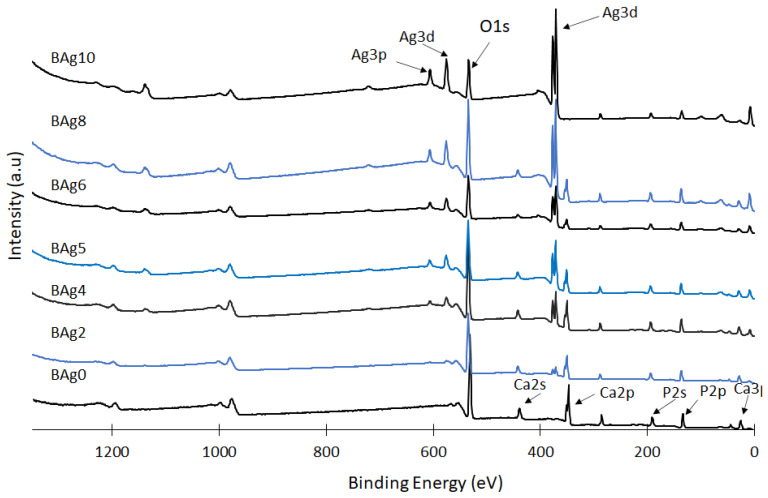
XPS spectra of the CaHPO_4_·2H_2_O and Ag_3_PO_4_ compounds.

**Figure 7 biomimetics-08-00547-f007:**
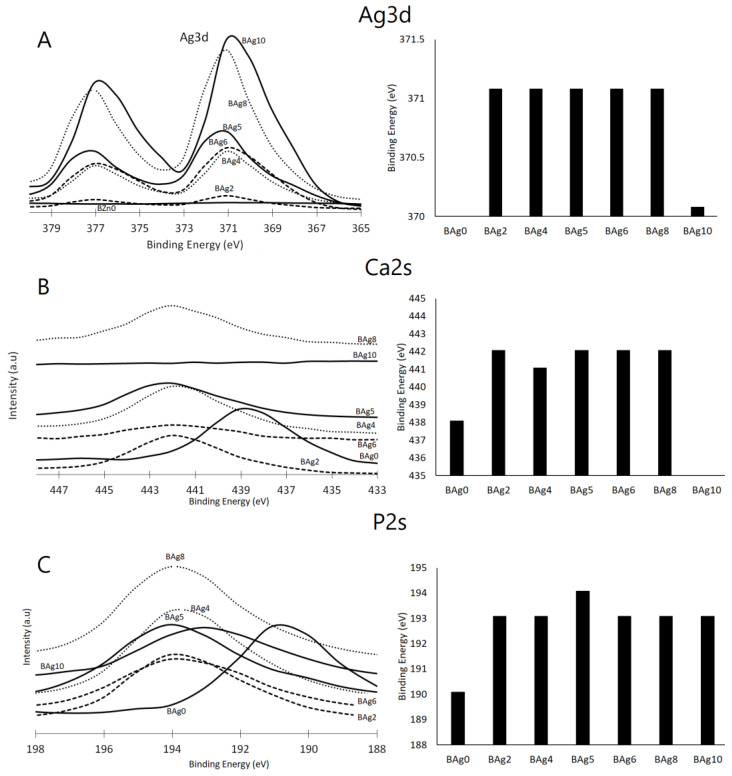
XPS analysis of the chemical state of the (**A**) Ca 2s, (**B**) P 2s, and (**C**) Ag 2p orbitals in the Ca_1−x_Ag_x_HPO_4_·nH_2_O compounds.

**Figure 8 biomimetics-08-00547-f008:**
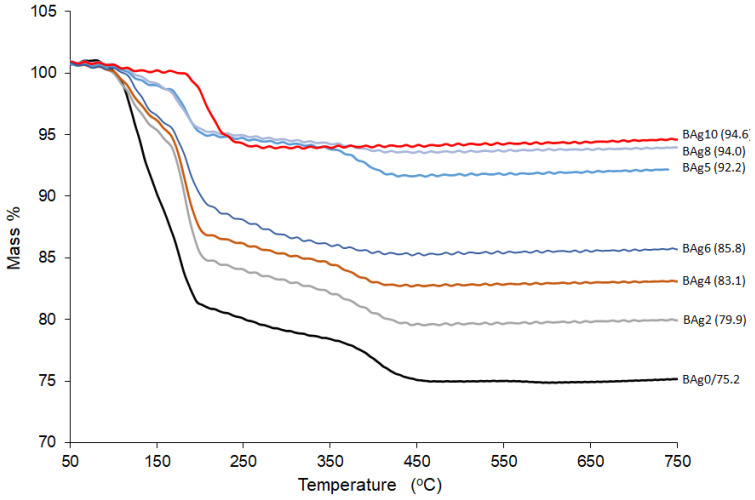
TG curves of the CaHPO_4_·2H_2_O and Ag_3_PO_4_ compounds (product names BAg0–BAg10).

**Figure 9 biomimetics-08-00547-f009:**
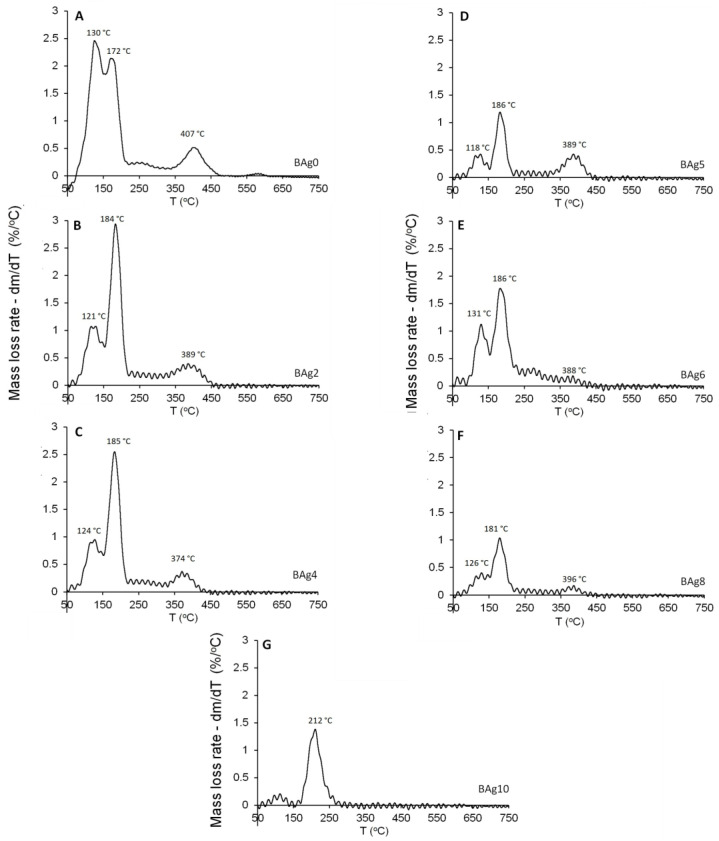
Differential TGA of the Ca_1−x_Ag_x_HPO_4_·nH_2_O compounds: (**A**) BAg0, (**B**) BAg2, (**C**) BAg4, (**D**) BAg5, (**E**) BAg6, (**F**) BAg8, and (**G**) BAg10.

**Table 1 biomimetics-08-00547-t001:** Molar proportions of (NH_4_)_2_HPO_4_, Ca(NO_3_)_2_·4H_2_O, and AgNO_3_, in addition to the Ag/Ca molar ratios applied for the synthesis of Ca_1−x_Ag_x_HPO_4_·nH_2_O compounds.

ID	(NH_4_)_2_HPO_4_	Ca(NO_3_)_2_·4H_2_O	AgNO_3_·6H_2_O	Ag/Ca Molar Ratio
BAg0	1	1	0	0
BAg2	1	0.8	0.2	0.25
BAg4	1	0.6	0.4	0.67
BAg5	1	0.5	0.5	1.0
BAg6	1	0.4	0.6	1.5
BAg8	1	0.2	0.8	4
BAg10	1	0	1	-

**Table 2 biomimetics-08-00547-t002:** The phase composition and parameters of the Ca_1−x_Ag_x_HPO_4_·nH_2_O unit cell, as determined by XRD scanning (Rietveld refinement, MATCH! software analysis). * Scherrer equation.

	Phase Composition	% wt.	Crystal Structure	a (Å)	b (Å)	c (Å)	ß°	Unit Cell Volume (Å^3^)	Crystallite Size (Å) *
BAg0	CaHPO_4_·2H_2_O	100	Monoclinic	5.82	15.22	6.27	116.41	496.65	53
BAg2	CaHPO_4_·2H_2_O	87.1	Monoclinic	5.81	15.19	6.24	116.41	493.43	50
	Ag_3_PO_4_	12.9	Cubic	6.00	6.00	6.00	90	216.43	21
BAg4	CaHPO_4_·2H_2_O	80.9	Monoclinic	5.81	15.18	6.24	116.43	492.91	65
	Ag_3_PO_4_	19.1	Cubic	6.01	6.01	6.01	6.01	217.03	33
BAg5	CaHPO_4_·2H_2_O	67	Monoclinic	5.81	15.21	6.26	116.41	495.65	56
	Ag_3_PO_4_	33	Cubic	6.01	6.01	6.01	6.01	217.03	34
BAg6	CaHPO_4_·2H_2_O	54.9	Monoclinic	5.81	15.21	6.26	116.41	495.65	70
	Ag_3_PO_4_	45.1	Cubic	6.00	6.00	6.00	90	216.00	30
BAg8	CaHPO_4_·2H_2_O	36.3	Monoclinic	5.82	15.22	6.27	116.41	496.65	130
	Ag_3_PO_4_	63.7	Cubic	6.00	6.00	6.00	90	216.00	18
BAg10	Ag_3_PO_4_	100	Cubic	6.00	6.00	6.00	6.00	216.00	40

## Data Availability

The data that support the findings of this study are available from the corresponding author (M. Alshaaer) upon reasonable request.

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
