# Peer review of "Preparation and Characterization of Mono- and Biphasic Ca1−xAgxHPO4·nH2O Compounds for Biomedical Applications"

_biomimetics, 2023, doi:10.3390/biomimetics8070547_

Round 1

Reviewer 1 Report

Comments and Suggestions for Authors

Report on the manuscript entitled: “

Preparation and Characterization of Mono- and Biphasic Ca-HPO4.2H2O/Ag3PO4 Compounds for Biomedical Applications

Manuscript ID: [Biomimetics] Manuscript ID: biomimetics-2683368

Authors: Fahad Abdulaziz, Khalil Issa, Mohammed Alyami, Satam Alotibi, Abdulaziz A. Alanazi, Abdel Mohaymen H. Taha, Asma M. E. Saad, Gehan A. Hammouda, Nagat Hamad, and Mazen Alshaaer  

Brushite (CaHPO4.2H2O) or dicalcium phosphate dihydrate (DCPD) compounds have received much attention as promising biomaterials which are likely to be used as bone substitutes because they can be absorbed and replaced by natural bone tissue over time, eliminating the need for additional surgeries. However, such biomaterials are vulnerable to bacterial colonization. To avoid this, metal ions including Ag+, Cu2+, and Zn2+ are used as antimicrobial agents in CaPs. Particularly, theses ions are used as replacement of Ca2+ in the structure of Brushite. The aim of this work is to replace Ca2+ ions with Ag+ ions in CaHPO4.2H2O and examine the as prepared materials using XRD, XPS, FTIR and TG, focusing on various CaHPO4.2H2O/Ag3PO4 biomaterials. We found that this work presents significant biomedical applications. However, since the manuscript is entitled “Preparation and Characterization of Mono- and Biphasic Ca-HPO4.2H2O/Ag3PO4 Compounds for Biomedical Applications”, some basic questions need to be clarified.

Basic questions to be clarified

1) In the whole introduction, we cannot see the novelty of this work. As the author explained in the manuscript, several studies have been carried out to explore the addition of Ag to CaPs. However, only one reference was provided [32], whereas more than one publication in the literature have addressed this question. In addition, the following issues are not elucidated:

ü  In the previous works which were carried out, what was the exact research question and what were the limits of the previous investigations?

ü  Compared to previous investigations, what is the novelty of the present work? How do the authors intend to solve the problem? If the previous investigations have already been addressed this question, what is the exact contribution of the present work?

2) As we already said, the work is about the preparation and the characterization of Brushite compounds obtained by replacing Ca2+ with Ag+. Unfortunately, in the introduction, there is no clear description of the structure of Brushite (Space Group, spatial arrangement of Ca layers with HPO4 groups, localization of water’s molecules, etc). We think that it is important to describe the structure and explain exactly how the replacement occurs in the structure, since Ca2+ is replaced with Ag+, in other words it is important to focus on the materials structure, preparation and characterization in the introduction. Unfortunately, the whole introduction was focused mostly on the medical aspects, which is in contradiction with their title “Preparation and Characterization of Mono- and Biphasic Ca-HPO4.2H2O/Ag3PO4 Compounds for Biomedical Applications”.

3) From the literature, it is known that:

(i) The presence of the DCPD phase as main phase was confirmed by the observed diffractions peaks.  However, a secondary phase was identified in increasing amounts for increasing Ag doping rates, which is especially detectable at 2θ = 29.70, 33.30° and 33.66°. This secondary phase, even present as traces could be identified as silver phosphate Ag3PO4.

(ii) It was also reported that an intimate mixture of the DCPD and Ag3PO4 phases was obtained in a one-pot synthesis, which could enhance existing cement formulations by allowing better/easier mixing of starting ingredients.

So my basic question is the following:  what are the main differences from what was reported to be known the literature and what has been described in the present work?

Specific questions

3) I don’t think that following notation Ca2HPO4·2H2O/Ag3PO4 is correct. In general such a notation with the sign “/” designates an interface between two phases, for example Si/SiO2 interface means that SiO2 layers are grown on top of Si and this leads to an interface between the two solid phases. Since, it is about an ionic substitution, maybe we should replace Ca with Ag in the formula.

4) Since the title contains the biomedical applications, why in vitro cell behavior experiences were not carried out with the as prepared biomaterials. What is it about?

General conclusion of the report:

We recommend the authors to elucidate these questions and clearly state the novelty of the present work. The authors should define the main research problem based on the limitations given in the literature. From these limitations, they should define what their exact contribution to solve the research question is.

Author Response

We thank the reviewer for her/his careful reading of the manuscript and her/his constructive remarks. We have taken the comments on board to improve and clarify the manuscript. Please find below a detailed point-by-point response to all comments.

1. In the whole introduction, we cannot see the novelty of this work. As the author explained in the manuscript, several studies have been carried out to explore the addition of Ag to CaPs. However, only one reference was provided [32], whereas more than one publication in the literature have addressed this question.

  • A paragraph related to the addition of Ag to CaPs was added, and several relevant references related were added. [Please see lines 100-122, and references: 36-40

2. In addition, the following issues are not elucidated:   In the previous works which were carried out, what was the exact research question and what were the limits of the previous investigations? Compared to previous investigations, what is the novelty of the present work? How do the authors intend to solve the problem? If the previous investigations have already been addressed this question, what is the exact contribution of the present work?

  • The gap in previous studies is addressed and novelty of this research is added and explained in the introduction, please see lines 133-147

3. As we already said, the work is about the preparation and the characterization of Brushite compounds obtained by replacing Ca2+ with Ag+. Unfortunately, in the introduction, there is no clear description of the structure of Brushite (Space Group, spatial arrangement of Ca layers with HPO4 groups, localization of water’s molecules, etc). We think that it is important to describe the structure and explain exactly how the replacement occurs in the structure, since Ca2+ is replaced with Ag+, in other words it is important to focus on the materials structure, preparation and characterization in the introduction. Unfortunately, the whole introduction was focused mostly on the medical aspects, which is in contradiction with their title “Preparation and Characterization of Mono- and Biphasic Ca-HPO4.2H2O/Ag3PO4 Compounds for Biomedical Applications”.

  • A paragraph related to the structure of brushite is added. Please see lines 75-85

5) From the literature, it is known that:

(i) The presence of the DCPD phase as main phase was confirmed by the observed diffractions peaks.  However, a secondary phase was identified in increasing amounts for increasing Ag doping rates, which is especially detectable at 2θ = 29.70, 33.30° and 33.66°. This secondary phase, even present as traces could be identified as silver phosphate Ag3PO4.

(ii) It was also reported that an intimate mixture of the DCPD and Ag3PO4 phases was obtained in a one-pot synthesis, which could enhance existing cement formulations by allowing better/easier mixing of starting ingredients.

  • Thanks for your valuable comments, which were taken into account in the abstract and conclusions. Please see lines 35-37 and lines 357-359.

6. I don’t think that following notation Ca2HPO4·2H2O/Ag3PO4 is correct. In general such a notation with the sign “/” designates an interface between two phases, for example Si/SiO2 interface means that SiO2 layers are grown on top of Si and this leads to an interface between the two solid phases. Since, it is about an ionic substitution, maybe we should replace Ca with Ag in the formula.

  • The notation Ca2HPO4·2H2O/Ag3PO4 was replaced by a chemical formula: CaxAg1-xHPO4.nH2O throughout the text and the title.7

7. Since the title contains the biomedical applications, why in vitro cell behavior experiences were not carried out with the as prepared biomaterials. What is it about?

  • This stage of research aims only to improve the synthesis of the precursors. The next stage will focus on the applications of these precursors including the in vitro characterization.

8. General conclusion of the report: We recommend the authors to elucidate these questions and clearly state the novelty of the present work. The authors should define the main research problem based on the limitations given in the literature. From these limitations, they should define what their exact contribution to solve the research question is.

  • Please see the changes in the abstract and conclusions beside the last paragraph in the introduction.

Reviewer 2 Report

Comments and Suggestions for Authors

1.      Title. Misprint in the chemical formula.

2.      Abstract should be expanded with 1-2 sentences to show for what it is good to synthesize the novel compounds described in the text.

3.      Introduction can be expanded with the information about the antibacterial properties of CaPs from the papers
Fadeeva, I. V., et al. "Tricalcium phosphate ceramics doped with silver, copper, zinc, and iron (III) ions in concentrations of less than 0.5 wt.% for bone tissue regeneration."
BioNanoScience 7 (2017): 434-438.
Deyneko, Dina V., et al. "Dependence of antimicrobial properties on site-selective arrangement and concentration of bioactive Cu2+ ions in tricalcium phosphate." Ceramics International 49.13 (2023): 21308-21323.

4.      Is there any suggestions which positions Ag ions occupy in the described compounds at low concentrations?

5.      All over the text there are many misprints with the upper and lower indices in the valencies, formulas etc. Please, correct!

Author Response

We thank the reviewer for her/his careful reading of the manuscript and her/his constructive remarks. We have taken the comments on board to improve and clarify the manuscript. Please find below a detailed point-by-point response to all comments.

1. Misprint in the chemical formula.

  • The notation Ca2HPO4·2H2O/Ag3PO4 was replaced by a chemical formula: CaxAg1-xHPO4.nH2O. All the other chemical formulas were reviewed and corrected.

2. Abstract should be expanded with 1-2 sentences to show for what it is good to synthesize the novel compounds described in the text.

  • Done, please see lines 33-37

3. Introduction can be expanded with the information about the antibacterial properties of CaPs from the papers
Fadeeva, I. V., et al. "Tricalcium phosphate ceramics doped with silver, copper, zinc, and iron (III) ions in concentrations of less than 0.5 wt.% for bone tissue regeneration." BioNanoScience7 (2017): 434-438.
Deyneko, Dina V., et al. "Dependence of antimicrobial properties on site-selective arrangement and concentration of bioactive Cu2+ ions in tricalcium phosphate." Ceramics International13 (2023): 21308-21323.

  1. Done

4. Is there any suggestions which positions Ag ions occupy in the described compounds at low concentrations?

  • There are several studies in literature about doping brushite with Ag small amounts such as https://doi.org/10.1016/j.mtchem.2019.100230.

5. All over the text there are many misprints with the upper and lower indices in the valencies, formulas etc. Please, correct!

  • The chemical formulas and the text were reviewed and corrected.
  • This version of the manuscript has been submitted to the MPDI editing services.

Reviewer 3 Report

Comments and Suggestions for Authors

The manuscript entitled (Preparation and Characterization of Mono- and Biphasic CaHPO4.2H2O/Ag3PO4 Compounds for Biomedical Applications), I found it an interesting work. However, there are some remarks have to be considered before recommending it for publication. 

1- Some complex sentences arise through the manuscript. Shorter and more comprehensive ones will be better.

2- The novelty of this work has to be expressed in the introduction section when compared to published corresponding studies.

3- In the XRD section, the presence of observed ammonium phosphate residues with different peak intensities is not clear. Please clarify this point.

4- In the XPS Fig. 6, no Ca ions were detected in BAg10 …. Further discussion is required.

5- The conclusion section has to be rewritten.

6- More recent and relevant references published in 2023 are recommended to be cited in this article.

Comments on the Quality of English Language

Some complex sentences arise through the manuscript. Shorter and more comprehensive ones will be better.

Author Response

We thank the reviewer for her/his careful reading of the manuscript and her/his constructive remarks. We have taken the comments on board to improve and clarify the manuscript. Please find below a detailed point-by-point response to all comments.

The manuscript entitled (Preparation and Characterization of Mono- and Biphasic CaHPO4.2H2O/Ag3PO4 Compounds for Biomedical Applications), I found it an interesting work. However, there are some remarks have to be considered before recommending it for publication. 

1. Some complex sentences arise through the manuscript. Shorter and more comprehensive ones will be better.

  • The text has been reviewed and corrected, and the manscript has was reviewed and corrected by the MDPI language editing.

2. The novelty of this work has to be expressed in the introduction section when compared to published corresponding studies.

  • The novelty and the previous studies were explained, please see lines 133-147

3. In the XRD section, the presence of observed ammonium phosphate residues with different peak intensities is not clear. Please clarify this point.

  • These are residual salts from the starting solutions. They should be washed out.

4. In the XPS Fig. 6, no Ca ions were detected in BAg10 …. Further discussion is required.

  • Done, please see line 291.

5. The conclusion section has to be rewritten.

  • Done

6- More recent and relevant references published in 2023 are recommended to be cited in this article.

  • Done: please see references: 32, and 36-40

7. Some complex sentences arise through the manuscript. Shorter and more comprehensive ones will be better.

  • The chemical formulas and the text were reviewed and corrected.
  • The final version of the manuscript was reviewed and corrected by the MPDI language editing services.

Reviewer 4 Report

Comments and Suggestions for Authors

Check carefull all formulae. Many miss subscripts etc. Avoid pompous wording. You may omit Fig. 1 since you descibe the syntheses procedures very thoroughly. Please study carefully the attached annotated manuscript and comply with the reviewer's suggestions.

Comments on the Quality of English Language

The authors tend to use pompous expressions that are inappropriate for a scientific text. The per is rather verbose and could be shortened.

Author Response

We thank the reviewer for her/his careful reading of the manuscript and her/his constructive remarks. We have taken the comments on board to improve and clarify the manuscript. Please find below a detailed point-by-point response to all comments.

1. Check carefull all formulae. Many miss subscripts etc. Avoid pompous wording. You may omit Fig. 1 since you descibe the syntheses procedures very thoroughly. Please study carefully the attached annotated manuscript and comply with the reviewer's suggestions.

  • The text was corrected according to the revision report provided by the reviewer.
  • All the chemical formulas were corrected.

2. The authors tend to use pompous expressions that are inappropriate for a scientific text. The per is rather verbose and could be shortened.

  • The final version of the manuscript was reviewed and corrected by the MPDI language editing services.

Round 2

Reviewer 1 Report

Comments and Suggestions for Authors

The authors have taken into account all my concerns. Additional references were added to show the novelty of their work. The revised version is suitable for a publication in the journal "Biomimetics". 

Author Response

Many Thanks!

Reviewer 2 Report

Comments and Suggestions for Authors

Authors took into account most of my concerns. But there is still misprints in the title and text (at least in my version of the manuscript) in the chemical formulae like

CaxAg1-xHPO4.nH2O

Please, check!

Author Response

Thank you so much for your review. We changed the  chemical formula to:

Ca1-xAgxHPO4.nH2O.